# Sequence specific detection of bacterial 23S ribosomal RNA by TLR13

Xiao-Dong Li[1], Zhijian J Chen[1,2]*

[1]Department of Molecular Biology, [2]Howard Hughes Medical Institute, UT Southwestern Medical School, Dallas, United States

**Abstract** Toll-like receptors (TLRs) detect microbial infections and trigger innate immune responses. Among vertebrate TLRs, the role of TLR13 and its ligand are unknown. Here we show that TLR13 detects the 23S ribosomal RNA of both gram-positive and gram-negative bacteria. A sequence containing 13 nucleotides near the active site of 23S rRNA ribozyme, which catalyzes peptide bond synthesis, was both necessary and sufficient to trigger TLR13-dependent interleukin-1β production. Single point mutations within this sequence destroyed the ability of the 23S rRNA to stimulate the TLR13 pathway. Knockout of TLR13 in mice abolished the induction of interleukin-1β and other cytokines by the 23S rRNA sequence. Thus, TLR13 detects bacterial RNA with exquisite sequence specificity.

## Introduction

Toll-like receptors are evolutionarily conserved transmembrane proteins that detect microbial components on the cell surface or within the endosomes (*Takeuchi and Akira, 2010*). All TLRs contain extracellular leucine-rich repeats (LRRs) and an intracellular Toll-interleukin-1 receptor (TIR) domain that recruits MyD88 and other adaptor proteins to activate signal transduction cascades, which culminate in the production of inflammatory cytokines and other antimicrobial molecules. Vertebrate TLRs comprise 6 major families, TLR1, TLR3, TLR4, TLR5, TLR7 and TLR11 (*Roach et al., 2005*). The TLR1 family includes TLR1, TLR2, TLR6, TLR10 and TLR14, which are localized on the plasma membrane. Within this family, TLR2 forms a heterodimer with other member of the family (e.g., TLR1, TLR6 or TLR10) to detect microbial lipopeptides and peptidoglycans. TLR4 and TLR5 also reside on the plasma membrane and detect bacterial lipopolysaccharide (LPS) and flagellin, respectively. TLR4, after binding to LPS, can also traffic to endosomal membrane where it launches a signaling cascade leading to the production of type-I interferons (IFNα and IFNβ). Members of the remaining TLR families, TLR3, TLR7 and TLR11 are localized on the endosomal membrane. TLR3 detects double-stranded RNA and induces inflammatory cytokines and interferons. The TLR7 subfamily consists of TLR7, TLR8 and TLR9. TLR7 and TLR8 detect single-stranded RNA whereas TLR9 binds unmethylated CpG DNA. The TLR11 family consists of TLR11-13 in mice and TLR21-23 in fish and frogs. TLR11 has been shown to recognize a profilin-like protein from the parasite *Toxoplasma gondii* and an unknown ligand from uropathogenic *E. coli*. The ligands and roles of TLR12 and TLR13 are unknown. To date, 10 TLRs (TLR1–10) have been identified in humans and 12 (TLR1–9, TLR11–13) in mice.

In addition to TLRs, the innate immune system in vertebrate animals consists of other microbial pattern recognition receptors, including RIG-I like receptors (RLRs), NOD-like receptors (NLRs) and C-type lectin receptors (CLRs). RLRs, which include RIG-I, MDA5 and LGP2, detect viral double-stranded RNA in the cytoplasm and activate a signaling cascade that leads to the production of type-I interferons and other antiviral molecules (*Yoneyama and Fujita, 2009*). Recently, we found that RNA from commensal bacteria could also stimulate the RIG-I pathway (*Li et al., 2011*). Mice lacking the mitochondrial protein MAVS (also known as IPS-1, VISA or CARDIF), which is an essential adaptor protein in the RLR

*For correspondence: zhijian.
chen@utsouthwestern.edu

**eLife digest** A central feature of the immune system is the ability to detect bacteria, viruses and other pathogens so that they can be repelled or neutralized before they cause lasting damage to an organism. Cells employ a number of different receptors that can detect these pathogens or the molecules they produce. Many of these are called pattern recognition receptors because they recognize certain signatures of microorganisms such as nucleic acids or carbohydrates. An important class of pattern recognition receptor is the toll-like receptor: there are many different families of the receptors, each recognizing a unique feature of bacteria or viruses. (The word toll, which means 'great' in German, refers to a gene whose mutations lead to striking phenotypes in flies, and has nothing to do with road and bridge tolls.)

Toll-like receptors have two parts that perform two different functions: when one part binds the relevant microbial molecules, the other part sends a signal that results in the production of effector proteins. These proteins include interleukin-1β, which helps to fight infection by causing the inflammation of tissue. To date, 12 different types of toll-like receptors have been found in mice, including three—known as TLR11, TLR12 and TLR13—that are not present in humans. Very little is known about the functions of TLR12 and TLR13. Humans, on the other hand, possess 10 different TLRs, only one of which, TLR10, is not found in mice.

Li and Chen have now discovered that TLR13 is responsible for detecting a certain type of ribosomal RNA called 23S ribosomal RNA that are present in bacteria but not in eukaryotic cells. Moreover, they have shown that a short sequence of 13 residues within the 23S ribosomal RNA triggers this pathway and leads to the production of interleukin-1β. The sequence of 13 residues is located at an active site in the RNA that catalyzes the synthesis of peptide bonds, and changing just one of these residues stops the production of interleukin-1β. Other forms of ribosomal RNA are unable to trigger the production of interleukin-1β. These results show that TLR13 differs from all other pattern recognition receptors because it is able to recognize a specific RNA sequence. Li and Chen went on to generate mice lacking TLR13 and showed that immune cells isolated from these mice failed to respond to bacterial RNA. These mice can be used to investigate the role of TLR13 in immune responses to bacterial infections in vivo.

pathway, are highly sensitive to experimental colitis in part because of defective immune response to commensal bacterial RNA. Interestingly, whereas IFNβ induction by commensal bacterial RNA depends on MAVS, the induction of proinflammatory cytokines, including interleukin-1β (IL-1β), depends on MyD88 (*Li et al., 2011*).

In the course of investigating the mechanism of MyD88-dependennt induction of IL-1β by bacterial RNA, we found that TLR13 was responsible for detecting the 23S ribosomal RNA of both gram-negative and gram-positive bacteria. Remarkably, a short sequence of 13 residues within domain V of 23S rRNA, which is known to be the catalytic center of peptide bond formation (*Nissen et al., 2000*), is both necessary and sufficient to trigger the TLR13 pathway. Point mutations within the sequence, herein termed ISR23 (Immune Stimulatory RNA of 23S rRNA), abolished the ability of the 23S rRNA to induce IL-1β. Mouse macrophages lacking TLR13 failed to induce IL-1β and other cytokines in response to ISR23. Thus, TLR13 is distinct from all other nucleic acid sensing pattern recognition receptors in that it recognizes a specific RNA sequence.

## Results

### Bacterial RNA induces IL-1β through a MyD88- and Unc93b1-dependent pathway

Our previous studies showed that transfection of RNA from *Lactobacillus salivarius* (LAB), a gram-positive commensal bacterium commonly found in the gastrointestinal tract, strongly induced IL-1β in mouse bone marrow derived macrophages (BMDM) and Raw264.7, a mouse macrophage cell line (*Li et al., 2011*). Interestingly, even when LAB RNA was added to the culture media of BMDM and Raw264.7 without the transfection reagent FuGENE, it still strongly induced IL-1β (*Figure 1A*), which suggests that the RNA detection probably does not involve a cytoplasmic RNA sensor. To determine

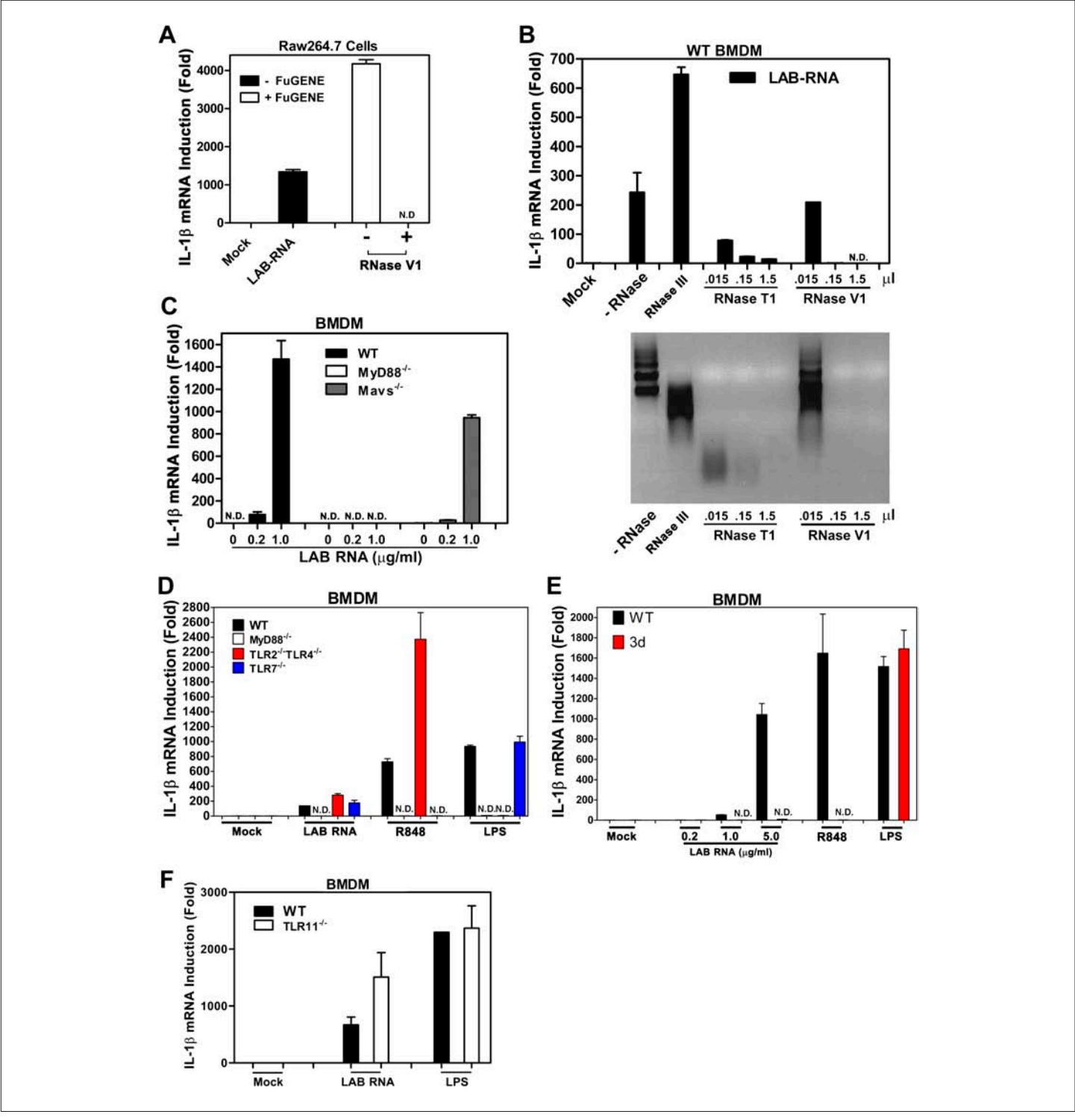

**Figure 1**. IL-1β induction by bacterial RNA depends on MyD88 and UNC93b1, but not MAVS, TLR2, TLR4 or TLR7. (**A**) *L. salivarius* total RNA (LAB RNA; 2 μg) was treated with or without RNase V1, then added to Raw264.7 cell culture in the presence or absence of FuGENE. IL-1β RNA was measured by qPCR. (**B**) LAB RNA was digested with indicated amounts of RNase III, RNase T1, RNase V1 or mock treated before adding to BMDM cell culture. 8 hr after incubation, total cell RNA was extracted to measure IL-1β expression by qPCR (upper panel). The efficiency of RNase treatment was verified by agarose gel electrophoresis (lower panel). (**C**) BMDM of the indicated genotypes was growing in the presence of LAB RNA at different concentrations for 8 hr, followed by the measurement of IL-1β RNA by qPCR. (**D**) BMDM of the indicated genotypes was growing in the presence of LAB RNA, R848 or LPS for 8 hr, then IL-1β induction was measured by qPCR. (**E**) Similar to (**D**), except that BMDM from Unc93b1 mutant mice (3d) was used. (**F**) BMDM from WT or TLR11⁻/⁻ mice was incubated with LAB RNA or LPS followed by measurement of IL-1β RNA by qPCR. Error bars represent standard error of triplicate assays. N.D: not detected.

what type of RNA was responsible for the activity, LAB RNA was treated with RNase III or RNase T1, which digests double-stranded (dsRNA) or single-stranded RNA (ssRNA), respectively. RNase T1 but not RNase III destroyed the IL-1β inducing activity of LAB RNA (*Figure 1B*). RNase V1, which digests both dsRNA and ssRNA at high concentrations, also destroyed the activity. Thus, ssRNA from LAB was responsible for IL-1β induction. This induction was abolished in BMDM from MyD88$^{-/-}$ mice but not Mavs$^{-/-}$, TLR2$^{-/-}$TLR4$^{-/-}$, or TLR7$^{-/-}$ mice (*Figure 1C,D*). To determine if the detection of LAB RNA occurs in the endosome, we used BMDM from the 3d mouse, which harbors a loss of function muta-tion in Unc93b1, a protein essential for the trafficking of endosomal TLRs from the ER to the endo-somal membrane (*Tabeta et al., 2006*). The induction of IL-1β by LAB RNA was abolished in the 3d BMDM (*Figure 1E*). As controls, IL-1β induction by the TLR7 ligand R848, but not the TLR4 ligand LPS, was dependent on Unc93b1.

## TLR13 is responsible for detection of bacterial RNA

Previous studies have suggested that members of the TLR11 family are localized on the endosomal membrane (*Brinkmann et al., 2007*; *Pifer et al., 2011*). Because IL-1β induction by LAB RNA depends on MyD88 and Unc93b1, but not other TLRs known to be involved in ssRNA detection, we investi-gated the role of TLR11 family members in detecting bacterial RNA. TLR11$^{-/-}$ BMDM induces IL-1β normally in response to LAB RNA (*Figure 1F*). To explore the role of TLR13, we constructed two lenti-viral shRNA vectors targeting distinct regions of TLR13 coding sequences and used the lentiviruses to generate Raw264.7 cell lines with stable knock down of TLR13 expression (*Figure 2A,D*). A lentiviral vector targeting GFP was used as a negative control. The knockdown of TLR13 by both shRNA vectors significantly reduced IL-1β induction by LAB RNA, but not by the TLR7 ligand R848 (*Figure 2A,B*). Importantly, expression of an RNAi-resistant TLR13 cDNA in the TLR13-shRNA cells rescued IL-1β induction by LAB RNA (*Figure 2C*). In fact, the IL-1β expression level in the TLR13-rescued cells was even higher than that in the WT cells, probably because of TLR13 overexpression (*Figure 2D*). These results indicate that TLR13 is required for IL-1β induction by the bacterial RNA.

## 23S ribosomal RNA induces IL-1β

To identify the ligand that activates the TLR13 pathway, we separated LAB RNA using formaldehyde denatured agarose gel. As the ribosomal RNAs are the dominant bands on the gel, we isolated these bands, extracted the RNAs and measured their activity. To avoid potential complications from other microbial ligands that might stimulate TLR2 or TLR4, we used TLR2$^{-/-}$TLR4$^{-/-}$ macrophages to measure IL-1β induction. Strikingly, 23S, but not 16S or 5S, rRNA potently stimulated IL-1β production (*Figure 3A*). This activity was not limited to gram-positive bacterial RNA, because 23S rRNA from the *E. coli* strain DH5α, a gram-negative bacterium, also strongly induced IL-1β (*Figure 3B*). To determine if the stark contrast in the IL-1β inducing activity of 23S vs 16S rRNA was due to their chemical modifications or their distinct sequences, we used T7 RNA polymerase to transcribe the *E. coli* rRNA in vitro (*Figure 3C*). Remarkably, the in vitro-transcribed 23S, but not 16S, rRNA induced IL-1β, indicating that 23S rRNA contains unique sequences capable of activating the TLR13 pathway.

## A short sequence within domain V of 23S rRNA stimulates the TLR13 pathway

Using in vitro transcription by T7 RNA polymerase, we carried out a systemic deletion analysis of 23S rRNA (*Figure 4*). Deletion of 520 nucleotides (nt) from the 3′ end (nt 1–2384 based on *E. coli* sequence) did not impair the ability of the RNA to induce IL-1β, but further deletion of another 490 nt (1–1894) did, suggesting that a sequence located between nt 1894 and 2384 is important (*Figure 4A,B*). Further deletion analysis narrowed down the stimulatory RNA sequence to nt 2035–2074 (*Figure 4C–H*), which was still fully capable of inducing IL-1β. To confirm and extend this result, we used chemically synthesized RNA corresponding to nt 2035–2074, 2035–2050 and 2054–2068 of 23S rRNA. This analy-sis showed that the 15 nt RNA sequence (2054–2068), ACGGAAAGACCCCGU, was a strong inducer of IL-1β (*Figure 5A*). Further deletion of one residue (A) from the 5′ end reduced the IL-1β stimulatory activity by about eightfold (*Figure 5B*; compare 2054–2068 with 2055–2068), and deletion of another residue (C) from the 5′ end abolished the activity (2056–2068). Deletion of two nucleotides (2054–2066) from the 3′ end was tolerable, but further deletion of another nucleotide (2054–2065) decreased the activity dramatically (*Figure 5B*). Titration experiments using varying concentrations of different RNA oligos suggest that the optimal immune stimulatory RNA sequence resides in 2054–2066 of 23S

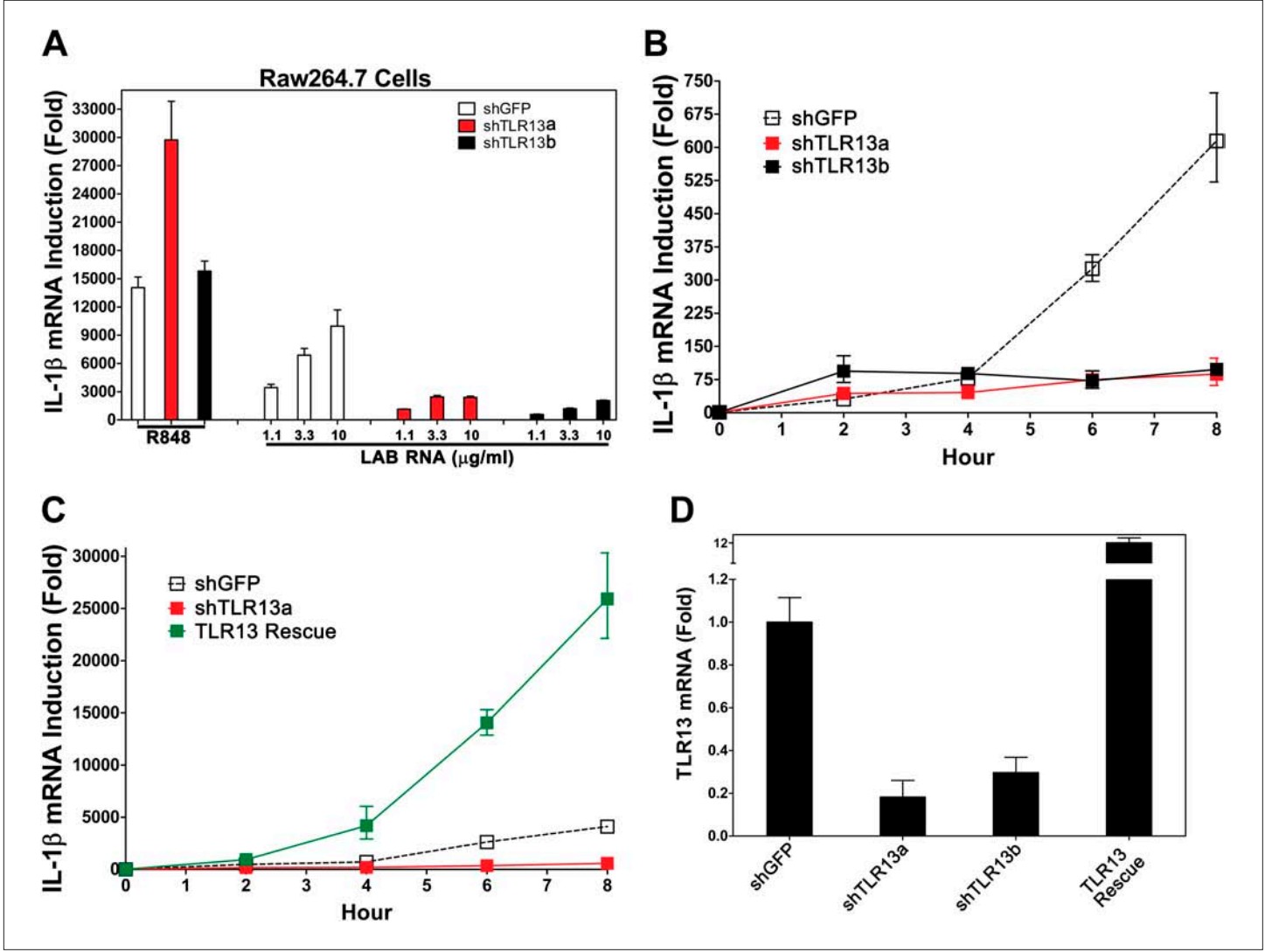

**Figure 2**. TLR13 is required for detection of bacterial RNA. (**A**) Raw264.7 cells stably expressing two distinct pairs of shRNA against TLR13 (TLR13a and TLR13b) or an shRNA against GFP (as a control) were growing in the presence of R848 (1 µg/ml) or LAB RNA at indicated concentrations for 8 hr. IL-1β induction was measured by qPCR. (**B**) Similar to (**A**) except that cells were growing in the presence of 2 µg LAB-RNA at the indicated times before harvest for qPCR analysis. (**C**) Similar to (**B**) except that an RNAi-resistant TLR13 cDNA was stably expressed in shTLR13a Raw264.7 cells. The TLR13 rescued cells were compared to shTLR13 and shGFP cells for IL-1β induction by LAB RNA. (**D**) The expression of TLR13 in the cells used in (**C**) was measured by qPCR.

rRNA (ACGGAAAGACCCC) (*Figure 5C*). For simplicity, we refer to this 13-nt sequence as ISR23 (Immune Stimulatory RNA from 23S rRNA).

To determine the sequence specificity of ISR23, we introduced point mutations in the sequence 2054–2068. Point mutations at each residue from position 2055 to 2064 abolished the activity of the RNA, whereas an A>G mutation at position 2054 was tolerable (*Figure 5D*). Notably, a mutation of A at position 2058 to any of the other three nucleotides abolished the activity, but 2'-O-methylation of A at this position had no detrimental effect (*Figure 5D*). We also introduced point mutations at positions 2058, 2060 and 2061 in the full-length 23S rRNA and found that each of these mutations abolished IL-1β induction (*Figure 5E*). The induction of IL-1β by both full-length 23S rRNA and nt 2054–2068 was markedly reduced by two distinct TLR13 shRNAs, indicating that ISR23 also engaged TLR13 to induce IL-1β (*Figure 5F*). Taken together, these results demonstrate that TLR13 detects the ISR23 sequence within 23S rRNA with exquisite sequence specificity.

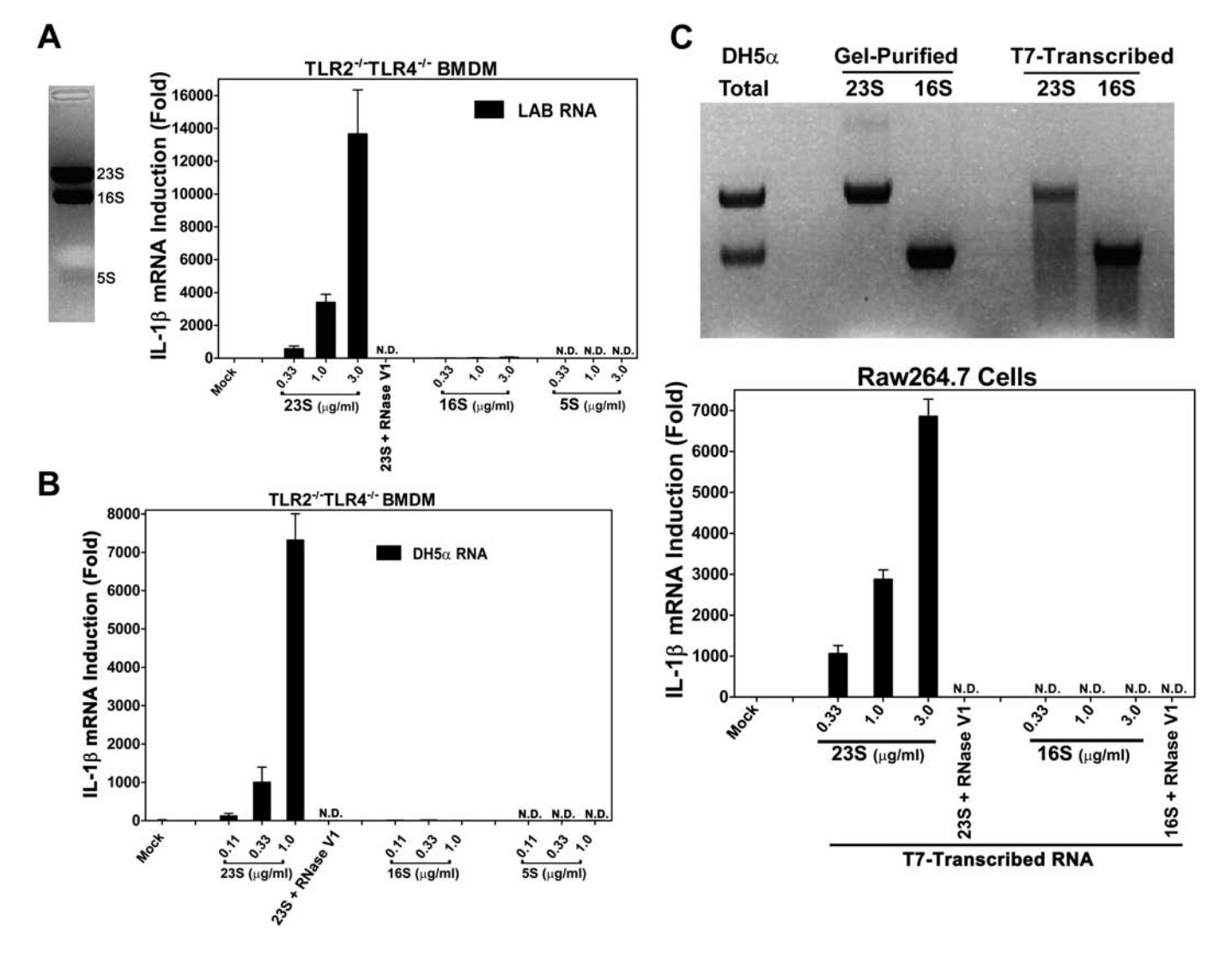

**Figure 3**. 23S rRNA stimulates IL-1β production in mouse macrophages. (**A**) LAB RNA was separated by denatured agarose gel electrophoresis (left panel) and ribosomal RNA was extracted. 2 μg of purified RNA was added to TLR2$^{-/-}$TLR4$^{-/-}$ BMDM culture and incubated for 8 hr. IL-1β mRNA expression was measured by qPCR (right panel). (**B**) Similar to (**A**) except using gel-purified rRNA from DH5α. (**C**) 23S and 16S DH5α rRNA was synthesized in vitro using T7 RNA polymerase and then gel purified. Indicated amounts of the purified RNA was added to Raw264.7 cell culture and incubated for 8 hr before IL-1β mRNA was measured by qPCR.

The ISR23 sequence is located in domain V of 23S rRNA, which is the catalytic center of the ribozyme responsible for peptide bond formation (**Cech, 2000**; **Nissen et al., 2000**) (**Figure 5G**; also see Discussion). The ISR23 sequence is highly conserved among gram-negative and gram-positive bacteria, including *E. coli*, *Salmonella*, *Listeria monocytogenes* and LAB (**Figure 5H**). However, *Haloarcula marismortui* 23S rRNA, for which the crystal structure has been solved (**Ban et al., 2000**), contains 4 nucleotides that diverge from the conserved sequence of ISR23. Interestingly, this ISR23 variant sequence from *H. marismortui* failed to stimulate IL-1β (**Figure 5H**). Thus, some bacteria might escape immune surveillance by TLR13 through mutations in the ISR23 sequence.

## Knockout of TLR13 in mouse macrophages abolished cytokine induction by 23S rRNA

We generated TLR13 knockout mice using a targeted ES cell line generated by the knockout mouse project (KOMP), which deleted almost the entire open reading frame in exon 2 and exon 3 of the

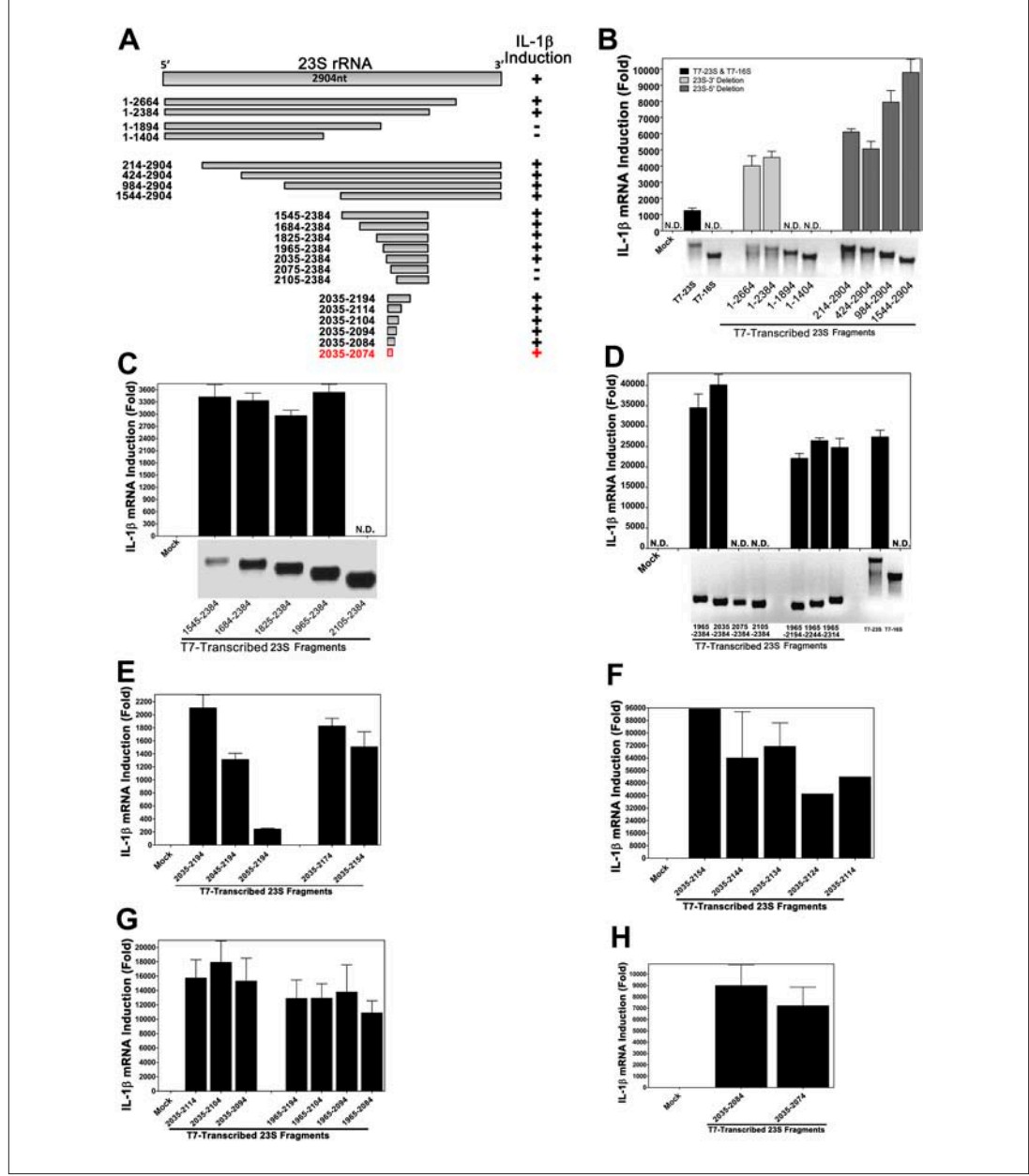

**Figure 4**. Structural and functional analysis of DH5α 23S rRNA. (**A**) Schematic summary of 23S rRNA deletion fragments and their IL-1β inducing activity. (**B**)–(**H**) DNA templates encoding full-length or truncation fragments of *E. coli* (DH5α) 23S rRNA were used for in vitro transcription using T7 RNA polymerase. The RNA products were purified and then incubated with Raw264.7 cells for 8 hr. IL-1β RNA levels were measured by qPCR.

mouse *Tlr13* locus on the X chromosome (*Figure 6A*). PCR of the mouse tail DNA and quantitative RT-PCR of spleen total RNA confirmed that the *Tlr13* gene was deleted in the knockout mice (*Figure 6B,C*). The Tlr13 KO mice were born and developed normally. BMDM derived from the WT and *Tlr13* KO mice were stimulated with the bacterial 23S rRNA sequence 2054–2068 or the TLR7 ligand R848 (*Figure 6D,E*). The deletion of *Tlr13* in the macrophages completely abolished IL-1β induction by the 23S rRNA sequence, but not by R848. To determine if stimulation of TLR13 triggers the secretion of mature IL-1β protein, which involves activation of the inflammasome, we immunoblotted bone marrow dendritic cell (BMDC) extracts and culture supernatants using an antibody against IL-1β. Stimulation of WT BMDC with the 23S rRNA sequence led to accumulation of pro-IL-1β, but not mature IL-1β (*Figure 6F*,

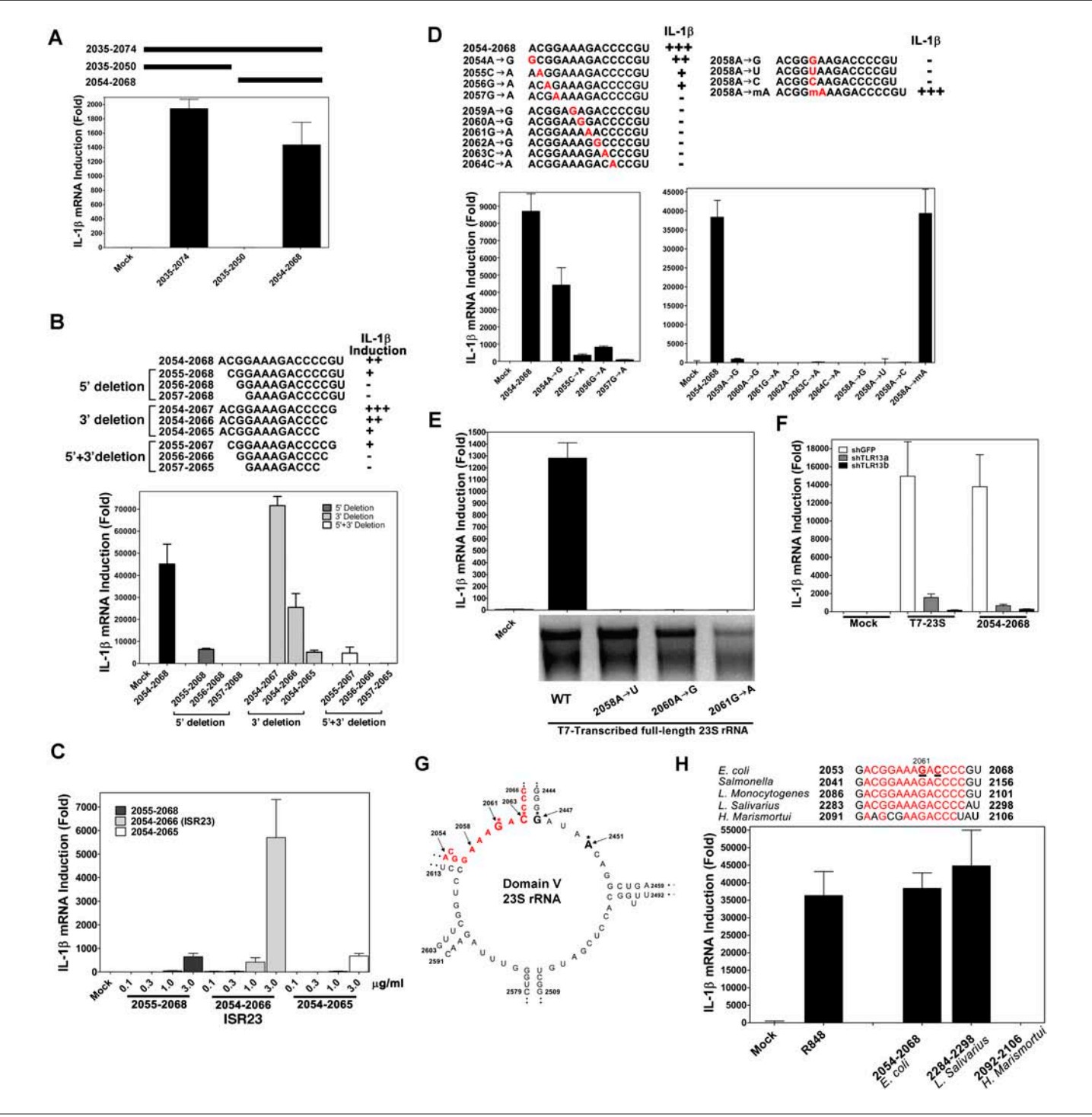

**Figure 5**. A specific sequence in domain V of 23S rRNA activates the TLR13 pathway. (**A**) and (**B**) Chemically synthesized RNA corresponding to the indicated region of 23S rRNA was added to Raw264.7 cells followed by measurement of IL-1β RNA by qPCR. (**C**) Similar to (**B**) except that different concentrations of the RNA oligos were tested for IL-1β induction. (**D**) RNA oligo corresponding to 2054–2068 of 23S rRNA and those containing the indicated mutations were tested for their ability to induce IL-1β. (**E**) Full-length 23S rRNA and that containing point mutations at the indicated positions were in vitro transcribed by T7 RNA polymerase and then measured for their ability to induce IL-1β in Raw264.7 cells. (**F**) Full-length 23S rRNA or the RNA oligo corresponding to 2054–2068 of 23S was added to Raw264.7 cell lines stably expressing shRNA against TLR13 or GFP. IL-1β induction was measured by qPCR. (**G**) Secondary structure of the domain V of *E. coli* 23S rRNA, with the ISR23 sequence highlighted in red. The invariant catalytic residues (G2061 and A2451) are shown in bold and indicated by an asterisk. (**H**) RNA oligos corresponding to the ISR23 sequence of different bacterial strains as indicated were added to Raw264.7 cell cultures followed by measurement of IL-1β by qPCR.

lane 5). When the cells were stimulated with the 23S rRNA sequence and then transfected with the DNA poly[dA:dT], which activates the AIM2 inflammasome (*Schattgen and Fitzgerald, 2011*), matured IL-1β was detected (lane 6). Neither pro-IL-1β nor mature IL-1β was detected in *Tlr13* KO cells stimulated by the 23S rRNA sequence (lanes 11 and 12). In contrast, LPS and poly[dA:dT] treatment induced pro-IL-1β and mature IL-1β even in the absence of TLR13 (lanes 9 and 10). These results indicated that TLR13 was essential for the induction of pro-IL-1β by bacterial ribosomal RNA but this receptor alone was insufficient to trigger inflammasome activation. TLR13-deficient macrophages were also completely defective in inducing other cytokines, including IL6, IL10, TNFα and MCP1, in response to the 23S rRNA sequence (*Figure 6G–J*). Similar results were obtained using macrophages derived from the spleen (data not shown).

## Discussion

Innate immune sensors invariably detect conserved microbial patterns that are indispensable for the life cycle of the microorganisms. Here we present evidence that TLR13 detects a highly conserved sequence at the catalytic center of the 23S ribosomal RNA of both gram-positive and gram-negative bacteria. Notably, G2061 (based on *E. coli* sequence) is hydrogen bonded to A2451, the catalytic residue of peptide bond synthesis (*Nissen et al., 2000*). Both G2061 and A2451 are completely conserved in the large ribosomal RNA subunits of all three kingdoms, suggesting a universal mechanism of peptide bond synthesis. Another residue of the ISR23 sequence, C2063, forms a base pair with G2447, which in turn forms a hydrogen bond with A2451 (*Nissen et al., 2000*). This hydrogen bonding increases the pKa of A2451, allowing it to serve as a general base to catalyze peptide bond synthesis. Remarkably, point mutations of several residues in ISR23, including G2061 and C2063, completely destroyed the ability of this RNA to induce IL-1β (*Figure 5D,E*). Thus, TLR13 targets the most conserved and essential feature of bacteria, namely peptide bond formation. This antibacterial mechanism of TLR13 is analogous to that of many antibiotics, which target the catalytic center of bacterial ribosomes (*McCusker and Fujimori, 2012*). Similar results have recently been published by *Oldenburg et al. (2012)*, who showed that the conserved 23S rRNA sequence 'CGGAAAGACC' is a ligand for TLR13. Through an independent and systemic analysis, we identified the optimal sequence that activated TLR13 as a 13-nucleotide sequence located in the active site of 23S rRNA ribozyme (ACGGAAAGACCCC; *Figure 5C*). Interestingly, Oldenburg showed that $N^6$ methylation at A2085 of *S. aureus* (corresponding to A2058 of *E. coli* 23S rRNA), which conferred antibiotic resistance, abolished TLR13 stimulation. Complementing these results, we showed that point mutations of the TLR13 recognition sequence in full-length 23S rRNA, including those expected to impair peptide bond synthesis, destroyed its ability to induce IL-1β (*Figure 5E*). Importantly, we have now provided the genetic evidence that knockout of TLR13 in mouse macrophages prevented the induction of IL-1β and other cytokines by 23S rRNA (*Figure 6*).

The extremely high degree of sequence specificity of TLR13 is unprecedented for nucleic acid sensing receptors. TLR3, TLR7, TLR8 and TLR9 detect RNA or DNA in the endosomal lumen without significant sequence specificity; instead, they recognize the structures of RNA or DNA (e.g., dsRNA for TLR3 and unmethylated CpG DNA for TLR9). RIG-I detects viral and bacterial RNA in the cytosol through the recognition of dsRNA bearing 5'-triphosphate. MDA5 detects viral dsRNA and perhaps some other unknown features. AIM2 binds dsDNA in the cytoplasm to trigger inflammasome activation. None of these cytosolic nucleic acid sensors display significant sequence specificity.

The high degree of RNA sequence specificity of TLR13 may allow rodents and other organisms that carry this gene to effectively combat some bacterial pathogens that threaten their survival while minimizing potential autoimmune attacks to the host. However, such sequence specificity also presents at least two liabilities to the innate immune system. First, although bacteria cannot mutate the invariant residues in 23S rRNA that are important for peptide bond synthesis (e.g., G2061 and C2063), they could mutate or modify other residues in the ISR23 sequences to render the bacterium invisible to TLR13, as shown for *H. marismortui* (*Figure 5H*). Second, the expansion of the mammalian genomes makes it possible that the exact 13-nt sequence of ISR23 may be found in some of these genomes, posing a threat of autoimmune reactions. Although mammalian ribosomal RNAs do not have significant sequence homology to ISR23, we found by a BLAST search that two human mRNAs, which encode the ribosomal subunit S14 (RPS14) and a pancreatic lipase (PNLIP), respectively, have the exact sequence match to ISR23. Since total RNA from human cells do not trigger

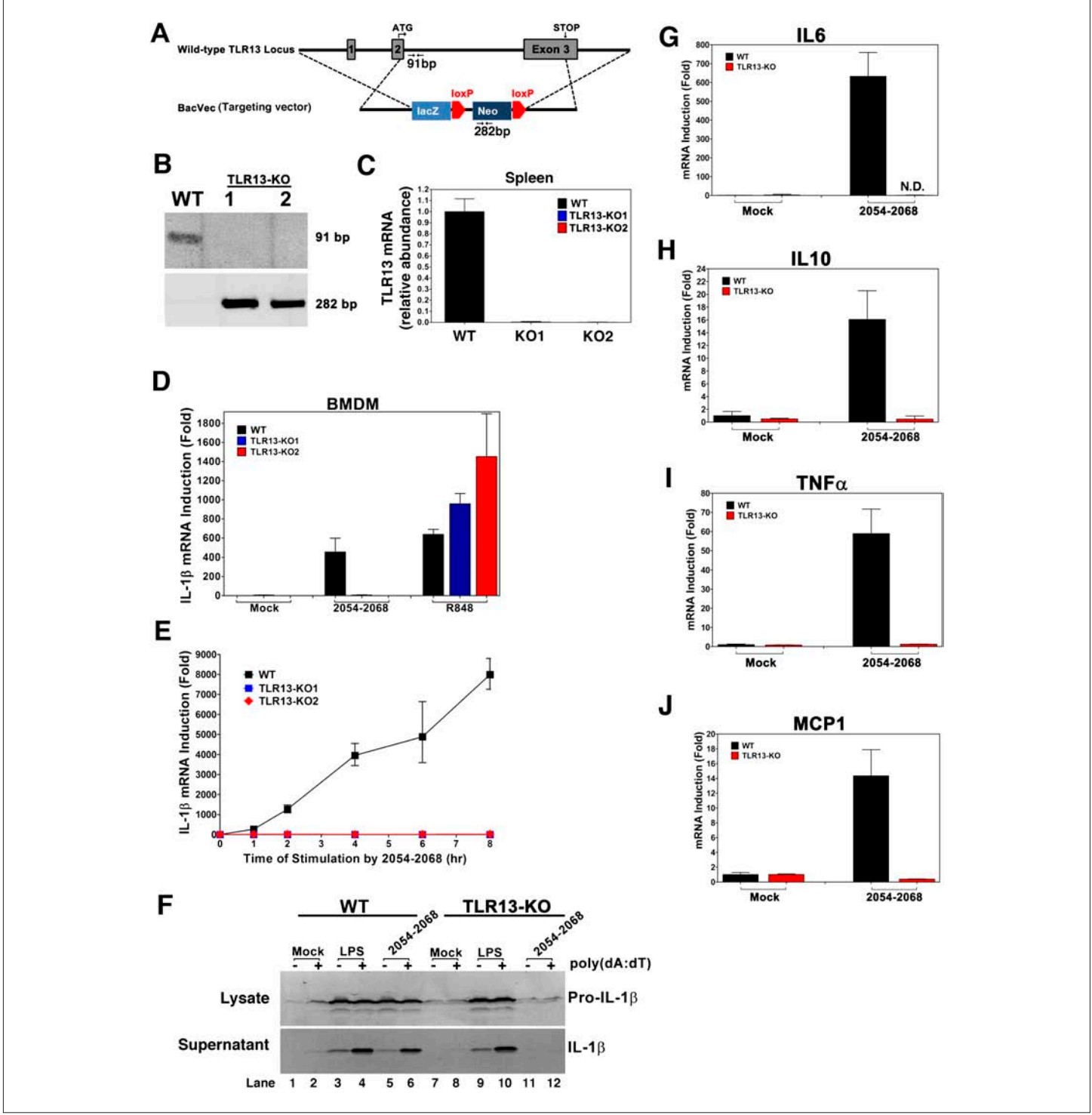

**Figure 6**. TLR13-deficient macrophages failed to induce cytokines in response to 23S rRNA. (**A**) Depiction of mouse *Tlr13* locus and gene targeting strategy. PCR primers and predicted sizes of the amplified fragments from WT and disrupted *Tlr13* loci are indicated. (**B**) Genotyping of one WT and two *Tlr13* knockout (KO) mice by PCR of tail genomic DNA. (**C**) qPCR of *Tlr13* RNA amplified from spleen total RNA. (**D**) BMDM from WT and *Tlr13* KO mice were incubated with the 23S rRNA sequence (2054-2068; 1 µg/ml) or R848 (1 µg/ml) and then total RNA was isolated for qPCR analyses of IL-1β. The results are representative of two independent experiments. (**E**) Similar to (**D**) except that BMDM was stimulated with the 23S rRNA sequence for different lengths of time as indicated. (**F**) WT and *Tlr13* KO BMDC were incubated with LPS (100 ng/ml) or the 23S rRNA sequence (1 µg/ml) for 8 hr followed by transfection in the presence or absence of poly[dA:dT] (1.5 µg/ml) for 5 hr. Cell lysates (upper) and culture supernatants (lower) were immunoblotted with an antibody against IL-1β. (**G**)–(**J**) BMDM from WT and *Tlr13* KO mice were stimulated with the 23S rRNA sequence and then the expression of the indicated cytokines was measured by qPCR. Error bars represent standard errors of triplicate assays. N.D: not detected.

IL-1β production in murine macrophages (data not shown), the ISR23-like sequence in these mRNAs may be folded into secondary structures that cannot be detected by TLR13. Nevertheless, it remains possible that unfolded RNA or fragments of these RNA (e.g., in dying or dead cells) containing the ISR23 sequence could stimulate TLR13. Thus, human might have abandoned TLR13 and relied on other pathogen receptors including RLRs, NLRs and other TLRs to detect pathogenic bacterial infections while avoiding autoimmune attacks. The loss of TLR13 in human might have also helped the expansion of the large commensal bacterial communities in the gut, which is important for the development of the immune system. In this regard, it might be interesting to determine whether there is a selective pressure against bacterial species that carry the ISR23 sequence in mice but not in human.

It is interesting to note that humans lack the entire TLR11 subfamily, including TLR11, TLR12 and TLR13 (*Roach et al., 2005*). TLR11 was shown to detect a profilin-like protein in *Toxoplasma gondii* and is important for the production of IL-12 in dendritic cells in mice (*Yarovinsky et al., 2005*). Despite the lack of TLR11 protein (due to a stop codon in the coding sequence), humans have highly effective innate and adaptive immune responses against *T. gondii*. As profilin is abundantly present in human cells, humans might have evolved to abandon TLR11 to avoid autoimmune attacks and rely on other innate immune sensors to detect the parasite infection (*Balenga, 2007*).

Despite its apparent absence in humans, the discovery of TLR13 as a sequence-specific bacterial RNA sensor offers an opportunity to study the role of this receptor in immune defense against bacterial infections. Our data obtained from the TLR13 knockout macrophages show that TLR13 is essential for cytokine induction by the bacterial 23S rRNA, a phenotype that has not been observed with other TLR mutant mice, including those lacking TLR2, 4, 7 or 11. However, this does not exclude the possibility that TLR13 may cooperate with other TLRs to detect bacterial infections. In fact, in the context of bacterial infections, multiple TLRs are likely to be activated by distinct ligands associated with bacteria, including peptidoglycans, flagellin, lipopolysaccharides, DNA and RNA. Thus, the relative contribution of TLR13 to immune response against pathogenic and commensal bacteria in vivo requires further investigation, which will now be facilitated by the availability of the TLR13 knockout mice.

How bacterial ribosomal RNA gains exposure to TLR13 is another interesting question for future research. Perhaps macrophages could take up bacterial particles through endocytosis and the lysis of bacteria in the endolysomal compartments expose the 23S rRNA and the ISR23-containing remnants to TLR13 on the endosomal membrane. Our finding that the ISR23 RNA could stimulate mouse macrophages without transfection suggests that TLR13 might take up bacterial RNA (i.e., from dead or lysed bacteria) on the cell surface and traffic to the endosome where it launches the signaling cascades. The TLR13 signaling cascade clearly engage MyD88 and Unc93b1, but the details of the signaling pathway requires further dissection. Finally, although ISR23 may not be an adjuvant for the development of human vaccines, its potent activity in stimulating cytokine production may be employed to boost the production of antibodies in vertebrate animals that possess the TLR13 pathway.

## Materials and methods

### Mice

TLR13 knockout mice were generated using ES cells produced by the KOMP Repository (UC Davis) (http://www.velocigene.com/komp/detail/10438). Two independently targeted ES cell clones were injected into Albino B6 blastocysts to produce chimeric mice, which were bred with Albino mice to obtain germline transmission. The heterozygous F1 progenies were intercrossed to obtain TLR13 knockout mice. For genotyping TLR13 knockout mice, tail genomic DNA was isolated from 7- to 10-day-old pups and then amplified by PCR using primers described in *Supplementary file 1A*. The PCR condition was: 95°C 3 min; 95°C 30 s; 60°C 30 s, 72°C 30 s, repeated for 35 cycles.

Wild type (WT), Mavs$^{-/-}$, Myd88$^{-/-}$, Myd88$^{-/-}$Mavs$^{-/-}$, Unc93b1 mutant (3d), TLR2$^{-/-}$TLR4$^{-/-}$ and TLR7$^{-/-}$ mice were bred and maintained under specific pathogen-free conditions in the animal care facility of University of Texas Southwestern Medical Center at Dallas. These strains were maintained on C57BL/6J background. All mice were engineered, housed and used according to the experimental protocols approved by the Institutional Animal Care and Use Committee.

## Preparation of bone marrow derived macrophages (BMDM) and dendritic cells (BMDC)

Bone marrow cells were collected from femurs and tibiae of mice. Cells were cultured in DMEM containing 10% fetal bovine serum (FBS), antibiotics, and conditional media from either L929 cell culture or GM-CSF producing cells. 24 hr later, non-adherent cells were transferred to a new plate and fresh conditional medium were added every other day up to the seventh day. Mature macrophages or DC were harvested and transferred to new plates for further experiments.

## Cell culture, stable cell lines and bacterial RNA stimulation

Murine macrophage cell line Raw264.7 was grown in DMEM supplemented with 10% FBS and antibiotics including penicillin, streptomycin and Normocin (InvivoGen, San Diego, USA). Stable TLR13 knockdown cells were generated using lentiviruses expressing shRNA against TLR13 and selected by puromycin (2 μg/ml). The efficiency of TLR13 knockdown or rescue in these stable cell lines was confirmed by either qRT-PCR or immunoblotting using FLAG (M2) antibody (Sigma-Aldrich, St Louis, USA). In all experiments except indicated otherwise, 1 or 3 μg bacterial RNA was added to the culture medium and incubated for 6–8 hr before cells were harvested for analyses.

## Isolation of bacterial RNA

*Lactobacillus salivarius* (LAB) (ATCC 11741) and DH5α were grown in Difco Lactobacilli MRS Broth or Luria Broth (LB) under either anaerobic chamber (BD GasPak, Franklin Lakes, USA) without shaking or normal aerobic with shaking at 250 rpm, respectively. After 16–18 hr, bacterial cells were collected into lysing Matrix B tubes (MP Biomedicals, Santa Ana, USA) and rapidly frozen down with liquid nitrogen. To isolate the total RNA, TriZol (Invitrogen, Grand Island, USA) was added and the tubes were vortexed at high speed with FastPrep (Thermo Electron Corporation, Waltham, USA) at 4°C. Crude RNA was further purified with RNeasy Mini Kit (Qiagen, Valencia, USA). RNA was treated with DNase I (Roche, South San Francisco, USA) (1 hr; 37°C) to remove potential DNA contamination.

## Extraction of bacterial RNA from native and denatured agarose gel

Total RNA isolated from either LAB or DH5α was separated by native or denatured 1% agarose gel electrophoresis. Each band containing ribosomal RNA was excised and soaked in RNase-free water overnight. On the following day, 5-butanol was used to remove extra water and then RNA was precipitated with 100% ethanol.

## In vitro transcription

MEGAscript T7 kit (Ambion, Austin, USA) was used for in vitro transcription according to manufacturer's instruction. DNA encoding full length bacterial 23S or 16S rRNA was obtained by PCR amplification from DH5α genomic DNA using Phusion DNA polymerase (Finnzyms, Waltham, USA) and subsequently cloned in pGEM-T vector (Promega, Madison, USA). DNA templates for in vitro transcription were amplified by PCR using primers listed in *Supplementary file 1B*. RNA oligos were synthesized by Sigma and listed in *Supplementary file 1C*. Point mutations of full-length 23S rRNA were carried out with Phusion Site-Directed Mutagenesis Kit (Finnzymes).

## DNA cloning and mutagenesis

Mouse TLR13 cDNA was purchased from InvivoGen and subcloned into a pTY lentiviral vector in which TLR13 was fused in frame with a C-terminal Flag tag. The primers for amplification of TLR13 is shown in *Supplementary file 1A*, so are the DNA oligos for construction of pTY lentiviral shRNA vectors that target distinct regions of TLR13 coding sequence.

## RNase treatment

Transfection of RNA into Raw264.7 or BMDM was carried out using FuGENE (Roche) or Lipofectamine 2000 (Invitrogen). For enzymatic treatments of nucleic acids, 1.0 μg of nucleic acids was treated with RNase III, RNase T1 or RNase V1 (Ambion) at 37°C for 1 hr. Enzyme-treated RNAs were purified with RNeasy Mini Kit (Qiagen) before adding to cell culture.

## Total RNA isolation and quantitative real-time PCR (q-RT-PCR)

To extract RNA, cells were first lysed in 1.0 ml of TRIzol (Invitrogen). Lysate was mixed with chloroform, and the aqueous phase was applied to RNeasy columns to obtain total RNA (Qiagen). The iScript cDNA synthesis kit (BioRad, Hercules, USA) was used to create cDNA from 0.15 µg of RNA. Quantitative RT-PCR was performed in Applied Biosystem Vii7 using SYBR Green and primers described in *Supplementary file 1D*.

## Acknowledgements

We thank Drs. Felix Yarovinsky and Reed Pifer for providing TLR11$^{-/-}$ bone marrow, Dr. Chandrashekhar Pasare and Wei Hu for TLR2/4 double knockout mice, Dr. Edward K. Wakeland for TLR7$^{-/-}$ mice and Drs. Lora V. Hooper and Breck A. Duerkop for *Lactobacillus salivarius*. We thank Dr. Lijun Sun, Jiaxi Wu, Haocheng Cai and Fenghe Du for helpful advice and assistance.

## Additional information

### Competing interests

ZJC: Reviewing Editor, *eLife*. The remaining authors have no competing interests to declare.

### Funding

| Funder | Grant reference number | Author |
| --- | --- | --- |
| Howard Hughes Medical Institute | 800635 | Zhijian J Chen |
| National Institutes of Health | R01 AI093967 | Xiao-Dong Li, Zhijian J Chen |

The funders had no role in study design, data collection and interpretation, or the decision to submit the work for publication.

### Author contributions

X-DL, Conception and design, Acquisition of data, Analysis and interpretation of data, Drafting or revising the article; ZJC, Conception and design, Analysis and interpretation of data, Drafting or revising the article.

### Ethics

Animal experimentation: All animal experiments were approved by the Institutional Animal Care and Use Committee of UT Southwestern and the approved animal protocol is APN 0825-07-04-1.

## Additional files

### Supplementary files

- Supplementary file 1. (**A**) DNA oligos used for TLR13 analyses. (**B**) DNA oligos for PCR amplification of DNA templates that direct in vitro transcription of 23S rRNA fragments. (**C**) RNA oligos used in this study. (**D**) DNA oligos for quantitative RT-PCR analyses of cytokines.

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
