## [Author Response]

*(1) The authors should mention, succinctly, the similarity and novelty of their current findings in respect to the paper that recently appeared in Science Express (19 July 2012 / http://www.sciencemag.org/content/337/6098/1111.abstract)*.

Oldenberg et al also reported the identification of a 10-nucleotide sequence (CGGAAAGACC) in bacterial 23S rRNA as a ligand for TLR13. Through a completely independent approach, we arrived at a similar conclusion. However, we found that the optimal sequence that activates 23S rRNA contains 13 nucleotides (ACGGAAAGACCCC), which we referred to as ISR23. We also provided direct evidence that point mutations in the ISR23 sequence in the context of full-length 23S rRNA abolished its activity (Figure 5E). Furthermore, we now provide genetic evidence that the knockout of TLR13 in mouse macrophages prevented the induction of IL-1β and other cytokines by the 23S rRNA sequence (Figure 6). We have added a paragraph in the paper to note the similar finding by Oldenberg et al.

*(2) Throughout the study, the only output examined for the TLR13-MyD88 signaling by 23S rRNA or ISR23 is the induction of IL-1β mRNA. It is most likely that this signal induces other genes such as those for TNF-α, IL6 and type I IFNs. It would be preferable that they show mRNA induction for at least one or two of these cytokines, so that the readers will not misinterpret that the TLR13-MyD88 pathway is specifically targeted to IL-1β gene in lieu of a broader response, assuming this is the case*.

We thank the reviewer for this suggestion and have performed the experiments with results now shown in Figure 6. Specifically, we showed that stimulation of macrophages by the 23S rRNA sequence induced multiple cytokines, including TNFα, IL-6, IL-10, MCP1 as well as IL-1β. Importantly, the induction of these cytokines was completely abolished by the TLR13 knockout, providing the definitive genetic evidence for the essential role of TLR13 in detecting bacterial 23S rRNA.

*(3) Is there any possibility that TLR13 signals in conjunction with other TLRs? It would be interesting to discuss this issue*.

We have added the following paragraph to the Discussion:

“Our data obtained from the TLR13 knockout macrophages show that TLR13 is essential for cytokine induction by the bacterial 23S rRNA, a phenotype that has not been observed with other TLR mutant mice, including those lacking TLR2, 4, 7 or 11. However, this does not exclude the possibility that TLR13 may cooperate with other TLRs to detect bacterial infections. In fact, in the context of bacterial infections, multiple TLRs are likely to be activated by distinct ligands associated with bacteria, including peptidoglycans, flagellin, lipopolysaccharides, DNA and RNA. Thus, the relative contribution of TLR13 to immune response against pathogenic and commensal bacteria in vivo requires further investigation, which will now be facilitated by the availability of the TLR13 knockout mice.”